# Increasing coverage and uptake of voluntary family planning in Uganda's emerging municipalities and secondary cities: An implementation research study protocol

**Rornald Muhumuza Kananura**[ORCID][1,2,3,4☯]*, **Catherine Birabwa**[1☯], **Jacquellyn Nambi Ssanyu**[1], **Felix Kizito**[5], **Alexander Kagaha**[6], **Sarah Namutanba**[5], **Moses Kyangwa**[5], **Othman Kakaire**[7‡], **Peter Waiswa**[1,2,3,5‡]

1 Department of Health Policy Planning and Management, Makerere University School of Public Health, Kampala, Uganda, 2 Center of Excellence for Maternal, Newborn and Child Health, Makerere University School of Public Health, Kampala, Uganda, 3 Advance Innovations for Transforming Health in Africa, Kampala, Uganda, 4 African Population and Health Research Center, Nairobi, Kenya, 5 Busoga Health Forum, Jinja, Uganda, 6 School of Public Health, University of the Witwatersrand, Johannesburg, South Africa, 7 Department of Obstetrics and Gynecology, Makerere University School of Medicine, College of Health Sciences, Kampala, Uganda

☯ These authors contributed equally to this work.
‡ OK and PW also contributed equally to this work.
* mk.rornald@musph.ac.ug

## Abstract

### Introduction

While urban areas are often perceived to have better access to healthcare services, including modern family planning (FP) services, urban dwellers including those with better socio-economic status are faced with multidimensional challenges that shape their access to appropriate FP services. In Uganda's urban spaces, there is currently a lack of understanding among service providers, civil society organizations, and individuals/communities regarding the implementation of interventions that promote informed choice and voluntary use of family planning services. This knowledge gap has profound implications for reproductive rights. This study seeks to enhance existing efforts towards increasing coverage and uptake of Voluntary Family Planning (VFP) in Jinja City and Iganga Municipality, central eastern Uganda. Our primary question is, "What interventions can effectively be packaged and delivered to increase the uptake of VFP among different segments of urban residents?"

### Methods

We propose to use the Human-Centered Design (HCD) approach to understand the needs and challenges of users and community capabilities in ensuring access to VFP services. Co-creating with stakeholders' engagement and a data-driven-centric approach will steer design and adaptation that respond to the different population segments within the urban space. As such, the study will be implemented in three phases: formative assessment, design and implementation, and implementation monitoring and evaluation. The

**Data Availability Statement:** No datasets were generated or analysed during the current study.

**Funding:** The implementation of the project is funded by John Templeton Foundation, Grant number 62045. The funders had no role in study design, data collection and analysis, decision to publish, or preparation of the manuscript.

**Competing interests:** The authors have declared that no competing interests exist.

**Abbreviations:** CHW, community health workers; FGD, Focus Group Discussion; FP, Family Planning; HCD, Human-Centred Design; HIP, high-impact practices; IDI, In-depth interviews; KI, Key informant interviews; SCM, supply chain management; SDPs, service delivery points; VFP, Voluntary Family Planning.

implementation process will incorporate robust monitoring, learning, and adaptation mechanisms. The primary focus of these mechanisms will be to utilize gathered information effectively to inform the design of the implementation and facilitate continuous learning throughout the process. The study will apply a process monitoring and evaluation approach to address questions related to what package of FP interventions work, for whom, under what circumstances and why.

## Discussion

Guided by strong learning and implementation flexibility, we hypothesize that our implementation will provide segmentation-specific high-impact interventions in an urban context.

## Registration

This implementation research protocol has been registered on the Open Science Framework (OSF) repository Registries (https://osf.io/vqxu9; DOI: 10.17605/OSF.IO/VQXU9).

## Introduction

Improving women's agency in deciding when and how to use family planning (FP) services is critical for their health and well-being [1–4], and is aligned with the Sustainable Development Goals (SDGs) [5, 6]. Access to appropriate FP methods is an impactful strategy for improving women's lives as it is a key mediator for most economic development outcomes [4, 6]. For instance, FP promotes the empowerment of women and adolescent girls by enabling them to complete their education, seize better economic opportunities, and fulfil their capabilities. Additionally, ensuring universal access to FP contributes to reduced maternal deaths by averting unwanted pregnancies [7]. Therefore, to achieve global and countries' SDGs reproductive health commitments, substantial investments in FP interventions within countries, regions and districts are still needed. Despite the increasing need for birth spacing and limiting in developing countries, FP services among the sexually active population are not universally accessed. For instance, contraceptive use in Africa is estimated at 36% compared to 65% in the rest of the world [8]. Similarly, the unmet need for FP in Africa is estimated at 22% compared to 15% in Oceania and below 10% in the rest of the world [8]. Furthermore, access to FP services among adolescents remains a challenge in developing countries despite the high rate of early sex debut (50%) and pregnancies (20%) in the region [9, 10]. For example, as of 2016, 50% of the 21 million pregnancies among adolescent girls (15–19 years) in developing countries were unintended (43% in Asia, 45% in Africa, and 74% in Latin America and the Caribbean) [9].

Like other African countries, FP use in Uganda remains low. For instance, contraceptive use among married women and the unmet need for FP among sexually active women are estimated at 39 and 32%, respectively [11]. However, access and utilization of FP services are affected by various demand and supply-side factors. On the supply-side, the persistent stock-out of FP commodities compounded with inadequate health providers' skills and other health system challenges continue to affect the provision, access, and utilization of quality FP services. The demand-side factors include inadequate awareness and knowledge of different FP options, the community perceptions of FP methods, trust in the health system, the cost of the services and limited services within the community [12, 13].

While urban areas are often perceived and reported to have better access to better health-care services [14], it is important to acknowledge that the population heterogeneity that is characterized by many forms of inequalities shapes access to appropriate healthcare services including FP [14, 15]. Some of the challenges that have been indicated to affect access to other health services such as lack of a primary healthcare structure, plurality of providers yet limited free quality primary healthcare and population dynamics [16] are critical in determining access to appropriate FP services. However, evidence of FP intervention packages that could be used to target the urban context in developing countries is elusive.

Uganda is undergoing increasing urbanization [17] but with limited additional infrastruc-ture, resources, and strategies to improve urban population well-being and health services including FP. While previously operating within the rural district health system framework, the emerging municipalities and secondary cities may present management, infrastructural and organization challenges that may affect service delivery including meeting the need for FP. While Uganda's Local Government Act highlights how cities are independent of the districts [18], to our knowledge the health planning and budgeting within emerging municipalities and secondary cities primarily focus on waste management with little focus on reproductive health including FP interventions. Although there is now a special consideration for urban health in the health sector and national development plans [19–21], the major challenge is how inter-ventions could be tailored to meet the needs of different groups of urban populations [21].

Over the years, several high-impact practices (HIPs) for FP such as community engagement and supply chain management have been developed to increase FP use (HIPS). However, there is a need for FP programs that are tailored to the needs, expectations, preferences, or lim-itations of target populations. While we are aware of several FP strategies such as the use of vouchers to increase postpartum FP and access in hard-to-reach areas [22], FP benefits cards among youths [23], social franchising [22], and community FP camps [24], these strategies are usually done with limited time to prepare the clients to properly decide on their own. More-over, incentivizing the community to access FP services is against the principles of VFP and could be considered as one way of FP coercion [4].

In Uganda's urban spaces, there is currently a lack of understanding among service provid-ers, civil society organizations, and individuals/communities regarding the implementation of interventions that promote informed choice and voluntary use of FP services. This has signifi-cant implications for reproductive rights, which can be understood in the context of the popu-lation's rights to reproduction [4]. Firstly, the lack of understanding hampers the fulfilment of the right to information. This leads to limited awareness about available FP methods, their benefits, and potential risks. Consequently, individuals are unable to make informed choices about their reproductive health due to a lack of accurate and comprehensive information. Sec-ondly, the lack of understanding creates barriers to the exercise of individual autonomy. With-out knowledge about interventions that promote informed choice and voluntary use of FP services, individuals may face obstacles in accessing the full range of contraceptive options they need. Additionally, inadequate understanding can result in a lack of appropriate counsel-ling and information, hindering individuals' ability to make informed decisions that align with their personal reproductive goals. Thirdly, the lack of understanding affects the availability and accessibility of FP services. Service providers may fail to offer a wide range of contracep-tive methods or may not effectively address the needs and concerns of diverse populations. This limited availability and accessibility restricts individuals' right to access the comprehen-sive and affordable FP services they require. Lastly, the lack of understanding perpetuates dis-crimination and stigma surrounding FP. Biases and misconceptions held by service providers, organizations, or communities contribute to the marginalization and judgment of individuals seeking FP services. This discrimination undermines individuals' rights to access VFP services

without facing discrimination or stigma. In this protocol paper, we showcase our co-designed package of strategies that will be implemented to increase the uptake of VFP services in two distinct urban settings of central-eastern Uganda: Iganga Municipality and Jinja City.

In line with the objectives of the Uganda Health Sector Development Plan and the second costed implementation plan for FP for Uganda [19, 21], we seek to augment and strengthen existing efforts towards increasing coverage and uptake of VFP in urban areas. Our primary question is, "What interventions can effectively be packaged and delivered to increase the uptake of VFP?"

## Theoretical framework

Day and Brown's, 1986 [25] FP Transaction Model (Fig 1), illuminates the intricate interactions among four pivotal elements: the prospective user, the potential provider, the spatial or perceptual gap that separates them, and the broader contextual backdrop in which the FP transaction unfolds. This conceptual framework offers a comprehensive lens through which we can empirically demonstrate the intricate path to FP access.

Within the realm of FP transactions, two central factors merit primary consideration in the utilization process: the potential user and the provider, representing the demand and supply facets, respectively. On the demand side, the model elucidates how the user's socioeconomic status, their perception of service quality, trust in the provider, and concerns regarding potential side effects all influence their journey towards the provider. On the supply side, the model delineates how the provider's infrastructure can effectively address these demand-side factors through the implementation of various strategies, capacity development, and the organization of healthcare facilities.

Moreover, this framework underscores the significance of discerning gaps that may obstruct or facilitate the transaction. It underscores the potential for accessing FP through an 'intervening provider,' or conversely, the existence of formidable barriers, such as substantial social or geographical distances that may prove insurmountable.

Crucially, the prevailing context or setting in which the transaction unfolds exerts a pivotal influence on whether it occurs or not [25]. In the specific context of Uganda, women may possess knowledge about the availability of FP options. However, the combination of difficulties in accessing their preferred methods and the influence of family authorities who desire to have many children often acts as a deterrent, preventing many women from actively seeking out and utilizing FP methods. While this model accentuates the importance of all four components of the FP transaction, it places even greater emphasis on their interplay and ongoing dynamics. This underscores the interconnectedness between the supply and demand sides of FP [25]. The selection of a particular FP provider hinges not solely on provider or user characteristics but rather on their intricate interplay, embedded within the transaction's context, collectively shaping the accessibility and dynamics of the FP transaction [25].

Drawing upon the foundational work of Day and Brown in 1986 (Fig 1), we employ this FP transaction framework as a cornerstone for developing our Theory of Change for implementation.

## Methods

### Study setting

The Urban Thrive project will be implemented in Jinja City and Iganga municipality, which are in the Busoga region, Eastern Uganda. Busoga region has one of the highest fertility rates in Uganda, and according to the 2016 Uganda demographic and health survey, 21% of adolescents aged 15–19 in the region have begun childbearing, 29% of married women are using

**Family planning transaction model**

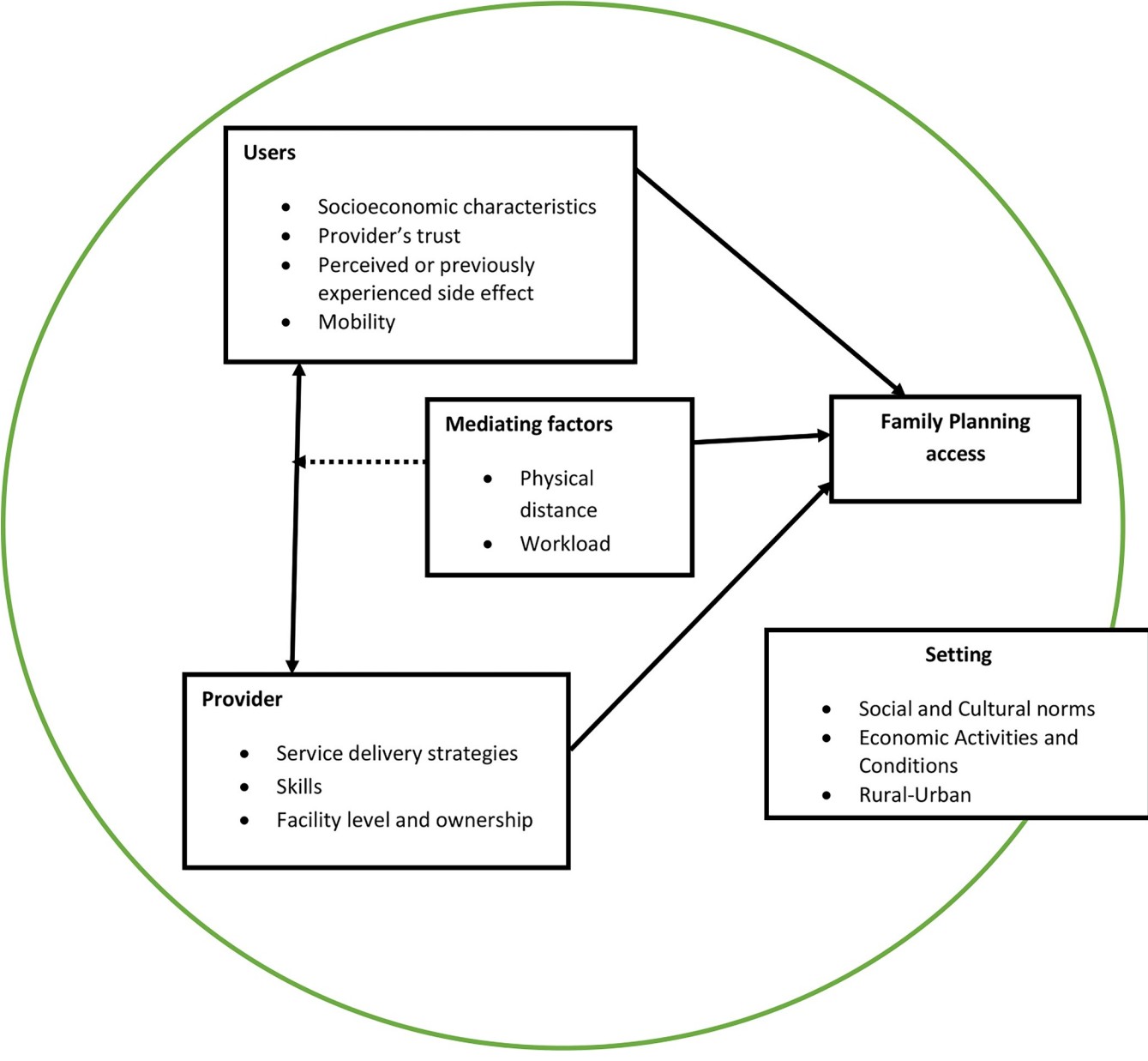

**Fig 1. Theoretical framework.**

modern contraception and 36.5% of married women have an unmet need for FP. The two areas combined have a population of at least 372,914 people living in urban centers. In Jinja alone, the night population is estimated at 307,414 and at 400,000 during the day. Jinja and Iganga districts are located along the Uganda-Mombasa (Kenya) transit route and host several commercial activities. The average population served per health unit is 15000, with 90% of the population living within a 5km radius of a health facility. Health service delivery is steered by

the city and municipal health officers. Like other upcountry cities and towns, there are ongoing FP promotion programs though still inadequate, leaving different population segments still in need of FP services.

## Study population

Our primary target populations are women and men of reproductive age (15–49 years for women and 15–65 for men) and health providers from the two study sites. We shall also engage city/municipality leaders and other key actors involved in reproductive health and community influencers including community leaders and civil society organizations. Additionally, we are cognizant of the influence of cultural and religious leaders, family members, and other population groups such as the motorcycle riders commonly known as boda-boda, whom we shall target during data collection, project design and implementation through behavioral change interventions.

## Implementation design

**Implementation research objectives and questions.** By integrating health systems approaches alongside established high-impact practices for FP [26–32], and ensuring comprehensive process documentation is an integral part of the project, the implementation efforts aim to shed light on how a customized set of interventions that address both demand and supply side issues can enhance the utilization of VFP services within an urban context. This approach will provide valuable insights into the effectiveness of tailored interventions in addressing the unique challenges associated with FP uptake in urban areas. Our implementation research objectives are:

i. To explore the current coverage of VFP and barriers and facilitators of contraception use in emerging municipalities and secondary cities settings in eastern Uganda;

ii. To improve understanding of FP and decision-making capacity for healthy reproductive or contraceptive behaviors and strengthen effective delivery and management of FP services through a tailored package of high-impact interventions that fit emerging urban settings in Uganda;

iii. To assess the effectiveness, facilitators, and barriers of the proposed intervention package in improving VFP service provision, accessibility, and utilization.

Our research questions are based on the phases of the project implementation: Formative, implementation and learning, and evaluation phases (Table 1).

**Theory of change.** A literature review informed the construction of the implementation theory of change (Fig 2) which clarifies the pathway to the implementation of the project. In the context of the study area, access to FP is shaped by a set of dynamic and complex factors that are ecologically interrelated [33, 34]. For instance, as indicated in Fig 1 (theoretical framework), some factors affect the initiation or non-use, continuation or discontinuation and misuse of contraceptives. To achieve the *a priori* implementation outcomes, our implementation approach applies various strategies in four domains: Social and behavioral change, service delivery, enabling environment and HIP enhancements (Fig 2).

*Social and behavioral changes*. At the community level, community leaders (cultural, religious) and social networks such as family members (husband, brothers, sisters, parents, and in-laws) influence the use of family planning [35–37]. Community-level authorities have control over resources, and emotional, social, and cultural structures.

**Table 1. Evaluation research questions.**

| Formative phase | Implementation and Evaluation phase |
|---|---|
| i. What are the met and unmet needs for FP in this urban population and what are the main underlying reasons? <br> a) Which population sub-groups in this urban setting are undeserved with FP services? <br> b) What are the factors associated with the discontinuation or non-use of contraceptives? <br> c) Which methods are acceptable and why? <br> d) What social norms, resources and networks influence voluntary contraceptive use in this setting? <br> Ii. What are the gaps and opportunities in the existing organization and delivery of FP services in these urban settings? <br> a) What is the coverage of FP services and methods in Iganga municipality and Jinja city? <br> b) What is the availability and quality of FP services and methods by sector and type of facility? <br> c) What high-impact practices for FP are being implemented in the two sites? What has been their effect on voluntary uptake? <br> d) What is the scale and scope of integration for FP services in the two sites? What are the effects so far of integrating FP services? To what extent is the community-based provision of FP services implemented? <br> e) What factors influence the organization and delivery of FP services? | Iii. What is the effect of community group engagement on contraceptive knowledge and behaviour among target groups? <br> iv. What is the feasibility, relevance and effectiveness of digital technologies for social and behavior change or improved service delivery? <br> v. What is the effect of the tailored package of HIPs on coverage or uptake of FP? <br> vi. What are the barriers and facilitators to accessing project interventions? <br> vii. What key aspects of the proposed package increase its adoption, penetration, and sustainability? <br> viii. What lessons can be identified that apply to other emerging cities and towns? |

In this project, we aim to increase the knowledge and understanding of VFP among women, men, and young people, which we assume will modify user perceptions and social norms about FP use, family size or birth spacing and will increase consistent and correct use. We hypothesize that if the community is engaged in discussions about the risks of not using FP, different FP options (expected side effects, costs, duration and utilization mechanisms), fertility awareness, and other desirable contraceptive behaviors; then their knowledge and understanding of risks, benefits and perceived barriers will be improved and this would increase individual and community acceptance of FP use, as well as increased autonomy of women and girls, to access FP services. The community groups' engagement, coupled with strengthened media communications and provider counselling, has been indicated to bring on a "peer effect" and stimulate women and families to proactively make appropriate contraceptive decisions that support or increase access to and use of VFP. Other various communication channels such as community groups, mass media, social media/networks and providers have been indicated as effective strategies in reaching the communities in resource-poor settings [34, 38]. Additionally, our knowledge-enhancing activities may also contribute to improved communication with providers and promote client-centered care.

*Service delivery*. We also presume that improving the delivery of VFP services will contribute to improved coverage, equitable access, and improved quality of FP services for all user sub-groups. This would contribute to increased utilization of FP. We hypothesize that building the capacity for the provision of quality VFP services at selected service delivery points and strengthening the availability of commodities through supporting commodity redistribution, while leveraging community-based systems, will contribute to increased access to a variety of appropriate FP choices and their correct use. Established and sustained well-functioning FP supply chains that meet the needs of public and private sectors, as well as health professionals

*Implementation theory of Change (Priori proposed interventions to increase uptake of family planning)*

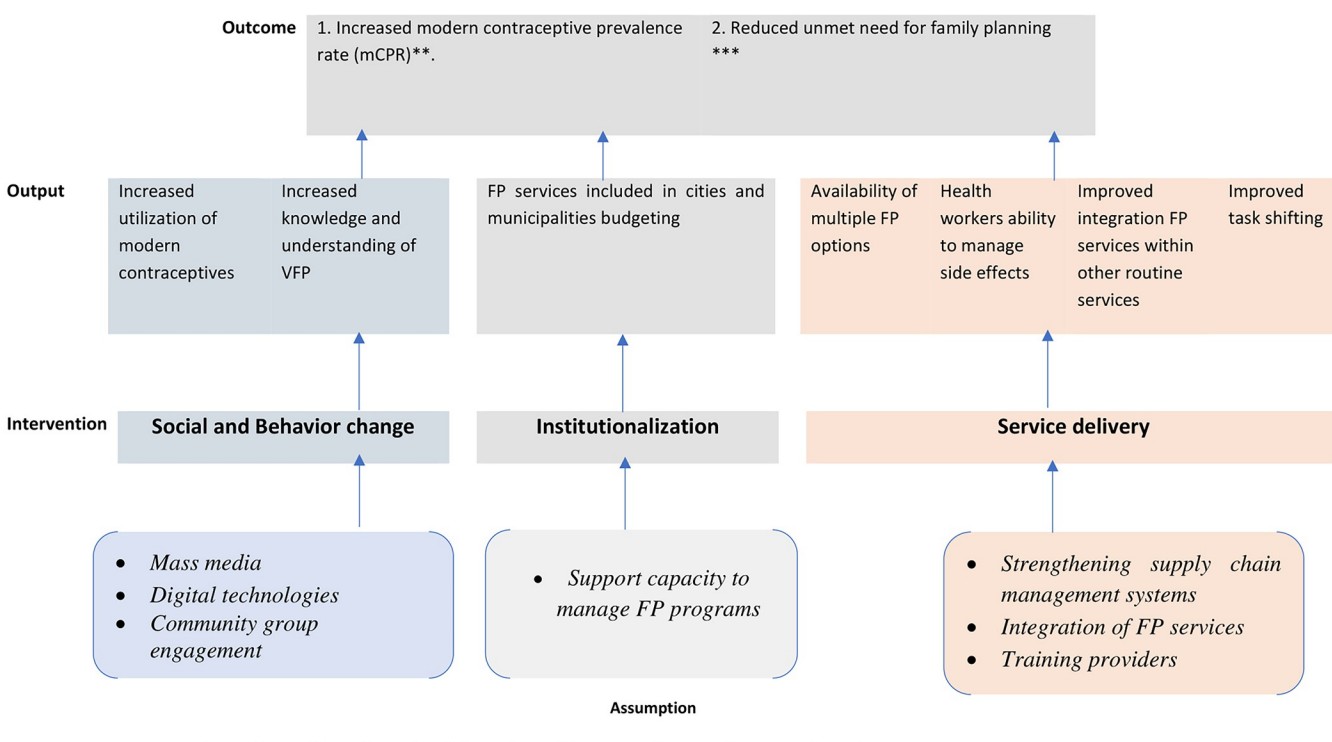

**Fig 2. Implementation theory of change.**

at all levels have been indicated to play a critical role in reducing the unmet need for modern contraception in developing countries [39, 40].

In this study, we shall design with the communities and health facility professionals to generate a package of strategies that will improve access to VFP services. Nevertheless, we are aware of some of the interventions that have worked in other settings in both the rural context of Uganda and other countries in the region. These strategies include working with community health workers to provide emergency contraceptives, working with private providers such as drug shops [33, 39, 40], and strengthening the health facilities to employ data-driven approaches for requisitioning a range of FP options that align with the specific demands of the target population.

We also hypothesize that supporting providers to include telehealth services in their packages will increase coverage of services given the diversity in urban populations and their health-seeking behaviors. Additionally, if referral care and clear channels for side effects management are put in place, users will feel better supported and motivated to continue VFP use.

*Governance and management.* Good governance and management are crucial in ensuring effective service delivery. If the alignment of the urban health system for FP is improved, services will be more responsive and acceptable, contributing to increased utilization. Also, if planning and integration of FP into urban plans and budgets are promoted, then commodity

security will be enhanced, resulting in reliable availability and continuous use among clients. Ultimately, with improved knowledge and understanding of VFP and improved service delivery and organization, coordinated by appropriate governance mechanisms, then the unmet need for FP will be reduced and voluntary use of FP will be increased.

As the current theory of change is based on our broader knowledge of FP access pathways, after the design and throughout the implementation phase, the theory of change will be reviewed and adapted where necessary based on emerging information. Furthermore, we are cognizant of several potential risks that could impact the successful execution of our implementation and as such, we have put forth comprehensive strategies to effectively mitigate these risks (S1 File).

**Implementation design and processes.** To refine and adapt the selected high-impact practices to the needs of the populations living in emerging urban spaces in Uganda, we propose to use the Human-Centered Design (HCD) approach [41, 42]. The HCD process focuses on understanding the needs, capabilities, and challenges of users; and then ensures stakeholder engagement in the development of solutions and design of programs through co-creation, inclusion, and transparency [41, 42]. Thus, HCD ensures that innovations match the users' culture, context, capabilities, opportunities, and constraints.

We will leverage the HCD principles to (1) identify any new ideas/solutions for addressing FP challenges in urban spaces; (2) to co-package high-impact interventions that are desirable, feasible and adaptable for these urban areas; and (3) to co-design implementation strategies to effectively deliver the selected interventions. Using the HCD approach will also enable us to better understand the contraceptive needs of users and the underlying enablers or barriers to VFP uptake. We will adopt three approaches of implementing the HCD as described by IDEO [41, 42], which are discovery, ideation, and prototype (Table 2).

The project activities have been segmented into three principal phases, as illustrated in Fig 3: Formative Assessment (comprising secondary data analysis, document review, and baseline data collection), Design, Testing, and Implementation, and Evaluation. It is worth noting that a continuous process of monitoring and learning has been seamlessly integrated into each of these phases.

In the context of the HCD framework, the alignment of these implementation phases is as follows: The Formative Phase and the ongoing monitoring during the Implementation Phase, corresponds to the 'Discovery'. The Design and Pre-testing Phase align with the 'Ideation', and the Testing and Implementation Phase is synonymous with the 'Prototype'.

**Table 2. Human-centered design process.**

| HCD process | Description |
|---|---|
| *Discovery* | The discovery process entails an in-depth exploration of both the end-users and the contextual factors surrounding a given problem. This will correspond to the formative study and involves sustained engagement with diverse stakeholders throughout the implementation of our proposed project. |
| *Ideation* | The ideation process is characterized by the generation of potential solutions aimed at addressing the identified problems. This creative process is facilitated through co-creation workshops and sustained interaction with various stakeholders during project implementation. |
| *Prototype* | The prototype process encompasses the development, testing, and refinement of prototypes. These prototypes pertain to both intervention packages and implementation strategies and are collaboratively crafted during co-creation workshops and ongoing engagement with diverse stakeholders during project implementation. During the implementation phase, they undergo field testing with potential beneficiaries, and any necessary refinements are made to ensure alignment with the needs of the target population and the contextual constraints. |

**Implementation phases**

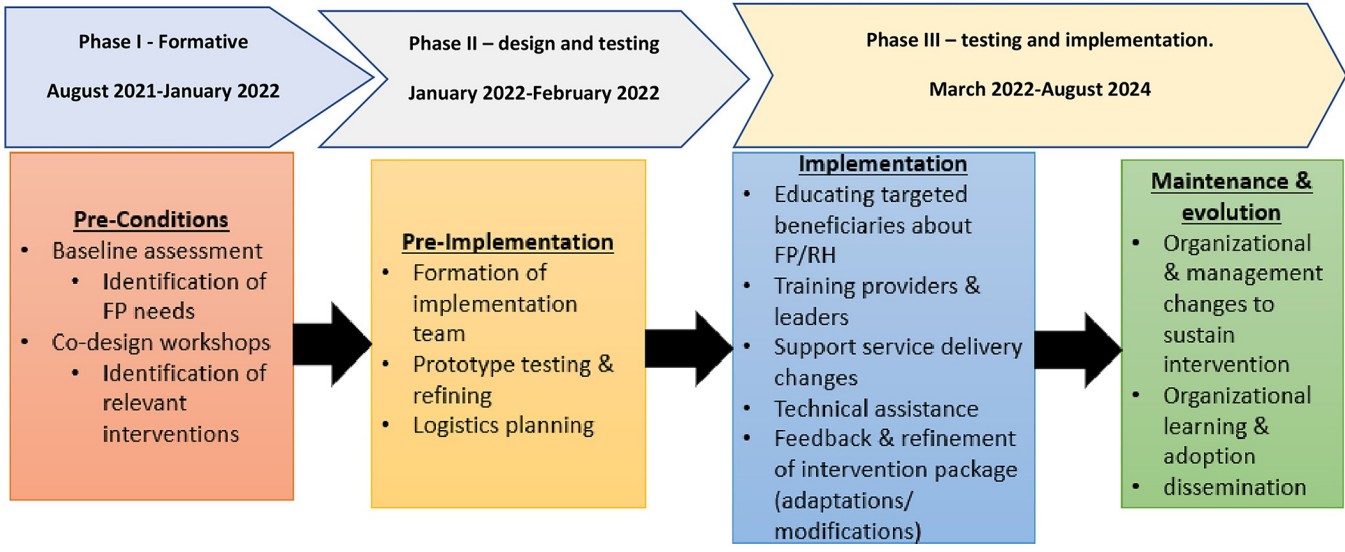

**Fig 3. Implementation phase.**

**Description of the proposed project interventions.** *Social and behavior change.* Despite a widely documented high prevalence of knowledge of FP methods, actual contraception use remains low in many settings [11, 43, 44]. This apparent contraception knowledge-behavior gap highlights potential inadequacies in the relevance of the information provided to users or failures in information interpretation/translation. A tailored health education program, based on community-identified needs, may help close knowledge gaps that still affect the uptake of FP. Our project seeks to enhance the quality of the information provided to users, to facilitate understanding and correct application of the knowledge acquired for VFP uptake. Table 3 summarizes the social and behavioral change activities and their respective targets.

*Group formation, training, and support.* Our approach to training and supervision of gender and age-sensitive participatory group is based on learnings from Tripathy et al., 2011 and

**Table 3. Social and behavioral change activities.**

| Activity | Description | Target |
|---|---|---|
| Training and supervision of gender and age-sensitive participatory community groups | The project will work with locally generated social groups to deliver targeted information on VFP to address key knowledge gaps, norms and other concerns that deter contraception use and promote healthy reproductive behaviors through group discussions and information sharing. | A total of 10 facilitators will be recruited and these will work in pairs. We target to form 10 groups in each site, each consisting of 40 members, giving a total of about 800 clients to be reached. The participatory learning and action cycles will be applied to three major groups (women, men, adolescents/youths). |
| Support and enhance implementation of media-based social and behavioral change and the use of digital technologies. | We leverage the existing technology such as WhatsApp and text messaging to reach different groups of people with information on family planning products and how they work including side effects. | We expect to mobilize about 1000 individuals per year, such that about 3000 will be reached over the project's life. We will create about 10 groups each of about 50 participants per year, making a total of 1,500 individuals. The number may increase due to innovation diffusion and increased willingness to join these social media communities. |
| Strengthen provider-initiated VFP counselling. | We shall build the capacity of the health workers to be able to integrate family planning communication or counselling in their routine work and across all service points. | Health providers will be supported to provide the information that matches the needs of the client, to facilitate uptake of VFP by ensuring relevant guidelines or standard operating procedures are available. |

Nahar et al., 2012 where community groups have been used successfully to change community norms and improve maternal and child survival [45, 46]. The proposed approach to community mobilization using facilitated participatory learning and action cycles with women's groups is a four-phase process, in which the groups collectively decide priority actions, guided by a trained facilitator. During the design and throughout the implementation we shall work with group members to identify and prioritize reproductive health or FP issues including perceived local barriers affecting their SRH and subsequently work with them to identify feasible strategies to address those issues.

In particular, we shall work with the groups in addressing issues of prohibited beliefs, stigma, how to reach vulnerable groups, manage side effects and how to link people to care. A series of group meetings on VFP will be conducted across the four phases. Facilitators will be expected to conduct meetings once every month, but in the first three months of implementation, the groups will meet twice a month. In addition, we will also target work niches (e.g., markets, boda boda stages, taxi parks), saving groups and other dominant social groups or community platforms. Care will be taken to ensure the representation of key socio-economic characteristics within each group, including occupation, education level, and marital status, among others. Group formation and identification of potential facilitators will be done in collaboration with relevant community representatives such as community health workers (CHW) or local chairpersons.

Before the commencement of the group formation and meetings, some preparatory activities will be done; for instance, during the formative phase, community mapping will be used to identify existing groups and influential individuals in the community. After the facilitators are recruited and trained, they will conduct some household visits to identify eligible individuals, explain the objectives of the groups and encourage them to participate in the group activities. This will be done in collaboration with the CHWs. Most of the meetings will be held with the community leaders to build rapport and gain their support for the group activities. Through the groups, women, men, and youths will be exposed to vast knowledge about FP, and if this is further discussed in the household, it may promote social understanding and cohesion on acceptable behaviors and probably increase social support for an individual's contraceptive or fertility choices. The groups will influence their local communities through community meetings. The participating community groups will be expected to hold community meetings to raise awareness of key reproductive health or FP problems, discuss their proposed strategies and provide feedback to the communities on the actions and progress. The groups will hold about 2 community meetings over the whole participatory learning and action cycle (one after phase 2 and one in phase 4). As such, the reach of the groups will be extended as the group members share knowledge with and provide support to other people in the community. The meeting sessions will be open to non-registered members, which will improve community acceptance, dissemination, and participation.

*Implementation of media-based social and behavioral change activities and the use of digital technologies*. Building on existing SBCC strategies, we shall develop tailored communication strategies that will be targeting priority sub-populations (such as, men, university and other eligible students, out-of-school young people, workers in bars and salons, among others.) or areas within the city. We will facilitate the airing of tailored radio messages through popular or preferred radio stations in the region and engage community radio stations to disseminate FP messages.

We also aim to strengthen or promote the use of mobile phone platforms to increase access to FP information and linkage to other resources, to the extent applicable based on population characteristics and contextual factors. This will include the generation and use of social media (WhatsApp groups and Facebook) communities/groups, leveraging social networks to

increase the spread of appropriate contraceptive knowledge and behaviors (social learning). Social media groups will particularly target youths and men, who have been shown to have higher ownership of smartphones. We will identify influential individuals, who will act as the group administrators to oversee overall group operations. These will also be expected to identify and invite potential eligible members to join the group. Each group will have at least one trained FP provider who will disseminate tailored FP messages and respond to any issues raised by the group members. We will use WhatsApp or Facebook based on utilization rates or acceptability/preferences of the target population.

The project will also utilize short message services to disseminate general FP information to mobile phone subscribers. Selected messages will be disseminated to different sub-populations/individuals that will be identified during the formative study and mobilized through the community groups and by the CHWs. One message will be sent per week over a period of 4–6 months. We will ensure fair representation of women, men, adolescents, young adults, and postpartum women. The content of the mobile phone-based messages will vary depending on gaps identified from the formative findings; but will include basic information on FP/RH, information on commonly used methods, guidance on initiation or safe switching of methods and reporting of side effects and sources for different options among other information. The duration and frequency of messages will be tailored to the targeted population groups, including the use of translated voice messages. We will also consider motivational messaging techniques or the use of interactive communication, where acceptable. We acknowledge that there will be variations in ownership and capacity to use mobile phones and related applications. Thus, strategies will be tailored to the population's characteristics.

We acknowledge that the use of digital technologies for social or behavior change for FP raises some ethical concerns like confidentiality or data security. These will be addressed at different project phases using various strategies such as anonymization, understanding pathways through which risks may arise and effective monitoring for real-time adaptation. We also plan to engage local drama groups to present skits on FP concepts in the communities. This will be supplemented with short videos for instance global health media videos [47]. Our multi-channel approach addresses some of the challenges posed by urban inequalities and the diversity of urban populations, needs and expectations; to increase exposure to FP information.

*Strengthen provider's initiated VFP counselling.* Family planning providers play a key role in educating and sensitizing users about VFP. However, inadequate access to appropriate FP services is also attributable to missed opportunities at the health facility or other points of contact with the health system. We will support/promote the integration of VFP counselling in key service points or special clinics at health facilities. This will be done through training and regularly reminding providers during support supervision. Emphasis will be put on women and men accessing general out-patient services, outreaches and women seeking maternal care, especially postpartum women.

**Service delivery.** Effective and efficient service delivery is vital to the attainment of universal access to sexual and reproductive healthcare services. The organization and delivery of FP services affect accessibility and continuity of care, especially among vulnerable populations, which in turn affects contraceptive choice and behaviors. Studies show a positive association between the quality of FP services and the use of modern contraceptives [48]. The project will support the provision of quality VFP services and equitable coverage through the understated activities. Table 4 summarizes service delivery activities and their respective targets.

*Development of knowledge and skills of healthcare providers.* Inadequate numbers or lack of trained providers remains a challenge in the provision of FP services [19, 49]. This affects the scope of methods provided and ultimately voluntary choice. As such, the project will work with the city reproductive health section and other implementing partners to establish training

**Table 4. Service delivery activities.**

| Activity | Target |
|---|---|
| Developing/enhancing knowledge and skills of healthcare providers | We target to train about 150 providers. The 150 providers from the selected health facilities will be expected to provide FP counselling and provide necessary information to all eligible individuals that seek care at their facilities. We anticipate working with 38 facilities, including HC IIs, HC IIIs, HC IVs and hospitals. |
| Strengthen availability of FP commodities and services | We target to engage the supply chain management officers at the sub-national level and the facility managers and stores managers from about 10 facilities in each site for this undertaking. The project will support or develop a system to collate and analyze relevant data to support decision-making by cities/towns relating to estimating supply needs or commodity distribution, as well as for process and performance improvement. |
| Strengthen community-based provision of VFP services | We propose to train 30 CHWs, 15 from each urban center. These will be expected to conduct monthly visits and discuss FP with at least 10 households, for a period of 6 months. |
| Strengthen application and use of digital technologies to support service delivery | Health facilities |
| Improve referral care and management of FP side effects | Health facilities and community health workers |

gaps for FP providers. This will be done through consultation, for example during co-design workshops and meetings for urban health authorities, as well as a review of existing implementation reports. This will help to prioritize which category of providers or places have the greatest capacity gaps for the provision of contraceptive methods, on which the project will focus and scale up provider training. The district health officer and city/municipality authority heads will lead the identification and selection of these providers/facilities and the list will be verified by the project's implementation advisory team. In addition, we will assess the readiness and commitment of the identified facilities to provide VFP.

Modes and level of engagement may vary based on the type of sector but overall, the project, through a community-based organization whom we will collaborate with, will develop a memorandum of understanding with the selected facilities and city leaders, who are responsible for planning and managing service delivery in the city and thus provide oversight. This will enable task-shifting for the provision of various methods, especially at places of initial care-seeking. A mixed approach will be adopted including "classroom" training, on-job mentorship, and simulation-based learning techniques. We will train the selected healthcare providers to enhance their knowledge, skills, and competencies in the provision of quality and respectful VFP services. We will use expert trainers and peer mentors, with regular supportive supervision. Mentoring will focus on strengthening the correct assessment of client FP needs and eligibility, client-centered method prescription, integrated service provision, respectful care, and youth-friendly care provision among others. The training of providers will adopt existing national FP training guidelines and manuals by the Ministry of Health and key implementing partners [50, 51]. Before the training, we will engage stakeholders such as partners that have previously trained providers in FP, and existing educators including national/local certified trainers, through consultative meetings or workshops to determine key areas/modules to focus on to improve the usefulness of the training.

*Strengthening availability of FP commodities and services.* Inadequate supplies of VFP commodities and resources continues to affect progress in reducing the unmet need for FP and women's ability to exercise voluntary choice of modern contraception. We will contribute to the improved availability of contraceptives or other VFP services by; 1) supporting capacity/

readiness strengthening initiatives in selected service delivery points (SDPs) and 2) improving supply chain management for FP commodities through increasing data visibility and use at all levels to inform inter-facility transfers or commodity redistribution.

In collaboration with the city leaders and partners, we will support readiness-strengthening initiatives at selected SDPs to ensure that they can provide quality services. We anticipate that there may be ongoing initiatives to increase the capacity of the facilities to provide quality FP services. This will be ascertained during our mapping exercises in the formative study. Findings will be discussed with the city leaders and other key actors to determine facilities or areas that are priorities for action. Depending on the SDP, activities will include training, service re-organization and building capacity for improved care coordination.

Furthermore, we propose to strengthen supply chain management (SCM) for FP commodities with a focus on increasing data visibility and use, as well as building the capacity of the supply chain workforce, especially at district and facility levels. We will first assess existing SCM capacity for FP across four main dimensions of supply chain organization: supply chain processes, technical infrastructure, and data capture and management as well as rational use. We will also identify factors contributing to SCM failures and any strategies towards logistics management improvement in the study sites during the formative study and use this information to identify key gaps and appropriate solutions to build on existing efforts. Our focus will be on data related to product movement within the city and town, using real-time approaches. This will involve analyzing routine data from SDPs on stock and consumption rates to gain insight into how much product is in inventory versus being regularly consumed. We will also participate in existing data quality improvement efforts.

*Strengthening the community-based provision of VFP services*. The project will support the community-based provision of VFP services by 1) strengthening the capacity of selected community health workers (CHWs) to correctly assess clients for FP, make appropriate method recommendations, disseminate gender-sensitive or age-appropriate information on FP or reproductive health, refer and follow-up of clients, as well as identifying and managing side effects among other basic FP services. The potential for CHWs to increase access and uptake of FP is widely documented [52, 53]. However, the implementation and effectiveness of the CHW model within urban areas may be undermined by various contextual factors including the mobility of the population [54]. These will be considered in the design of strategies that will be adopted to strengthen the CHW model in these urban areas. Our focus will be on training, promoting the integration of CHWs into the Primary Health Care system and strengthening support mechanisms.

Emphasis will be put on the provision of adolescent/youth-friendly FP services and effective male engagement. We will support the integration of CHWs by improving linkages between service delivery points and CHWs as well as households/families. This will involve ensuring that CHWs receive regular support supervision from the facilities they are attached to and encourage the utilization of information generated by the CHWs through their reports. We will leverage the existing CHW networks operating in the two sites, to strengthen their capacity to provide quality basic FP services and linkage of clients to more skilled services when needed. The CHWs will be expected to record their activities in a register or pre-designed activity logbook, which will be shared with the health facility to which they are attached and with the project team. We will assess the reach and impact of these CHWs using the following measures: number of women, men and youths reached by the CHWs, number of women, men and youths sensitized about RH and FP, number of women, men and youths referred to health facilities for FP services by CHWs, and number of clients who obtain FP methods from the CHWs.

Furthermore, we also aim to strengthen the community health system that best supports access and utilization of VFP in these urban settings. The community health system comprises CHWs, community groups, an operational management committee and the formal health system [55–57]. Adopting this conceptualization, this activity will be linked to the community groups' activity under the Social and Behavior change component, and we will also leverage structures that are already in place to increase information dissemination and linking to other VFP services. This includes education institutions (teachers), places of worship (religious leaders), and recreational centers (local chairpersons, 'local celebrities') among others. We propose to use these as platforms to educate communities about reproductive health issues within other prevailing/emerging social issues. Some might be used as service linkage points. The information provided will be tailored to each platform.

The engagement of religious leaders will be informed and built on previous efforts within the region. Some religious leaders and all the major religious-based health institutions have previously been trained and engaged to promote fertility awareness among the people and the importance of having manageable families. The leaders would mostly promote methods that are acceptable within the religion but would also create awareness about the availability of other options. We, therefore, plan to learn from this approach of engaging religious leaders and we will also identify leaders that have previously been trained and engaged in FP activities. We will build on the change norms among religious institutions to promote fertility awareness and healthy reproductive behaviors. We will also borrow these strategies and experiences to expand the influence of social norms on providers and the community.

*Strengthening application and use of digital technologies to support service delivery*. The ongoing COVID-19 has had far-reaching effects on the provision, access, and utilization of reproductive and other essential services [58]. For continuity of services both now and in future preparedness, promoting the institutionalization of digital technologies is necessary [59]. The project will identify existing providers of or systems for virtual FP services and strengthen implementation of virtual services like eligibility screening, counselling, shared decision-making, reporting commodity status, support supervision and coordinating providers. During the formative study, we will explore existing capacity and preference for digital interventions and infrastructure. This will guide the necessary interventions. Activities that will be undertaken include training of providers and sensitizing the public.

*Improving referral care and management of FP side effects*. Fear or experience of side effects is one of the major reasons for the non-use or discontinuation of contraception. We will sensitize and educate users and providers on the correct identification and reporting of side effects associated with FP use. We will train providers in the proper management of common side effects and other response mechanisms to ensure that users are supported to sustain the correct use of different contraceptive methods. We will work with the city leaders to ensure that providers have the necessary guidelines and standard operating procedures to support them in handling side effects and other complaints related to FP use. We will also identify a sustainable hierarchy or network of skilled providers, including gynecologists, doctors, or selected midwives, that can be easily accessed by clients.

*Governance and management*. Developing and/or supporting the capacity to lead and manage FP programs is a high-impact practice that helps to create an enabling environment by improving the work climate, improving organizational and management systems, and creating the capacity to respond to change (High-Impact Practices in Family Planning, 2015). Table 5 summarizes governance and management activities and their respective targets.

*Improving alignment of VFP services to reduce unmet need*. We will examine data on the availability of VFP services and client care-seeking patterns and identify gaps in the alignment of services that can be targeted to improve the efficiency of VFP service delivery. This evidence

**Table 5. Governance and management activities.**

| Activity | Target |
| --- | --- |
| Improve alignment of VFP services to reduce unmet need | City and municipality council members |
| Institutionalization and sustainability: Support better planning and integration of VFP into urban plans | City and municipality council members |

will be shared with the city/town leaders, to inform decisions on improving the organization and management of VFP services. The project will collaborate with/support the leaders to develop evidence-based plans and designs for VFP services to ensure that services are where the clients are and minimize missed opportunities.

*Institutionalization and sustainability*: *Support better planning and integration of VFP into urban plans.* We will strengthen capacity and support urban health leaders and health managers to develop evidence-based plans to guide VFP service delivery and harness opportunities for integrating VFP into existing health and other social services. This will be done by supporting strategic planning for health, annual planning, and quarterly review meetings for the two urban health authorities. Also, through our interactions with the facilities, we hope to support data quality improvement initiatives to ensure that quality urban health data is available to inform decision-making.

## Monitoring and evaluation design

We shall apply the process monitoring and evaluation approach to address questions related to what package of FP interventions works, for whom, under what circumstances and why [60]. Guided by the project theory of change (Fig 2) and the indicators with their measurements (Fig 4), we will apply mixed-method and multimethod approaches to data collection. Qualitative approaches will be linked for the exploration of the mechanisms or causation in FP access.

Different approaches to quantitative methods will also be applied to understand VFP access. Participant observations will be documented while in the community and during stakeholder meetings and interviews. We shall then document and reflect on certain participants' attitudes toward the topics or questions under discussion, such as silence, gesture, feelings, and stereotypical sentiments. The main aim of this approach will be to quantify the fidelity and how the package of our FP interventions was delivered; quantify FP uptake, discontinuation and switching clients' demographics; and qualitatively explore how our interventions improved the quality and reach of FP services.

**Process monitoring.** We shall generate a structured template that will be used to guide the reflection discussions on the significant contribution of the project, implementation facilitators, implementation barriers and possible solutions for accelerating progress. The template will be aligned with the implementation components and will focus on each component's activities. The template will include 1) the objective and activity being implemented; 2) the implementation process; 3) what is working well and why; 4) what is not working well and why; 5) what has changed or what have we stopped doing; and 6) what should we do next. The purpose of these reflections will be to document key activities, challenges, and adaptations occurring over the course of implementation.

Furthermore, during the implementation phase, we will actively involve diverse stakeholders in a concerted effort to comprehend the multifaceted barriers to accessing FP services among various population segments. Our primary emphasis will be on the collection and documentation of information regarding the perceived utility of various FP options, their

## Indicators and their measurements

| Results | Indicators and their measurements |
|---|---|
| **Outcome** | |
| 1. Increased Modern Contraceptive Prevalence Rate (mCPR) | **Indicator:** The percentage of sexually active women of reproductive age are currently or were using modern contraceptive methods in the last 12 months. **Measurement:** Calculated as the number of individuals using modern contraceptives divided by the total sexually active population within a specific time frame. The following are considered modern methods: female sterilization, male sterilization, Intrauterine Device (IUD), injectable, implants, pill, emergency contraception, male condom, female condom, other vaginal methods (foam, jellies/spermicide, diaphragm) Standard Days Method (SDM), Lactational Amenorrhea Method (LAM). |
| 2. Reduced Unmet Need for Family Planning | **Indicator:** The percentage of women in need of FP services who are currently or were not using any contraceptive method in the last 12 months. **Measurement:** Include women of reproductive age who want no more children or to postpone having the next child but are not using a contraceptive method. We also include women who are currently or were using a traditional method of family planning and women who are pregnant with or postpartum amenorrhea after an unintended pregnancy. |
| **Output** | |
| 1. Availability of Multiple FP Options | **Indicator:** The diversity and accessibility of contraceptive methods offered within the program. **Measurement:** A comprehensive inventory of available FP methods, including details on accessibility, affordability, and user acceptability. |
| 2. Health Workers' Ability to Manage Side Effects | **Indicator:** The ability of healthcare providers in addressing and managing side effects associated with FP methods. **Measurement:** A qualitative assessment involving healthcare provider interviews, skills evaluations, and clients' feedback to gauge their competence in providing effective care. |
| 3. Improved Integration of FP Services within Other Routine Services | **Indicator:** The extent to which FP services are seamlessly integrated into existing healthcare services. **Measurement:** A comprehensive analysis of integration efforts within the routine healthcare services including outpatient, immunization days, community outreaches, and maternity. |
| 4. Improved Task Shifting | **Indicator:** The effectiveness of redistributing specific FP-related tasks among healthcare providers including community health workers to optimize service delivery. **Measurement:** Health facility assessment will include documentation of task shifting including the healthcare cadres involved, and the impact on service efficiency. |
| 5. Inclusion of FP Services in Cities and Municipalities Budgeting | **Indicator**: The allocation of financial resources within the budgets of cities and municipalities specifically earmarked for FP services. **Measurement:** A qualitative interview with municipalities and cities' technical personnel to assess the extent of financial commitment to FP services by local governments. |
| 6. Increased Utilization of Modern Contraceptives | **Indicator:** The number of individuals utilizing modern contraceptive methods within the program's target population. **Measurement:** A comparison of baseline and current health facility utilization data will be used to assess the changes in the use modern contraceptives. |
| 7. Increased Knowledge and Understanding of Voluntary Family Planning | **Indicator:** The enhancement of knowledge and comprehension of Voluntary Family Planning principles and options among the users, health workers and policy makers. **Measurement:** The Voluntary Family Planning principles' measure include Knowledge of different FP options and how they work including side effects; availability of various options of FP in health facilities that are affordable to the users; availability of skilled personnel to administer and terminate FP option based on clients choice; health workers ability to address FP side effects; perception of the clients' on health workers attitudes towards client's side effects concern; perception of the clients' on health workers attitudes towards client's fertility; and women autonomy in accessing FP; |

**Fig 4. Indicators and their measurements.**

accessibility, the quality-of-care services provided, and the overall perception of FP within the community.

**Qualitative data.** The qualitative data collection will include key informant interviews (KIIs), in-depth interviews (IDIs) and focus group discussions (FGDs). Using a key informant interview guide, we shall conduct KIIs with urban health officials, service providers and implementing partners. We target 20 KIIs, 10 in each site. Using an in-depth interview guide, we shall hold interviews with women and young people to gain deeper insight into the dynamics of FP decision-making experiences and major barriers to contraception use. About 60 IDIs will be conducted, 30 from each site. The FGD interview guide will be used while holding interviews with separate groups of women, men, and young people. A total of 12 FGDs of 9 members will be conducted, giving a total of about 108 participants. The final sample size will depend on when data saturation is achieved. Among other issues, interviews will help us identify the main social norms that drive observed contraceptive behaviors. These will inform community group engagements and the strengthening of the community health system. We will include endline interviews or one-on-one dialogue meetings with potential clients after the implementation of the community engagement activities and media information dissemination.

**Quantitative data.** *Household, place of work listing and individual surveys*. The sampling frame will consist of all villages with their respective population. For programming purposes, we shall use Lot Quality Assurance Sampling (LQAS) techniques for data collection and interpretation of results. All the participating municipality divisions in the two districts where the

intervention is to be implemented will be selected (2 in Iganga and 3 in Jinja). We shall randomly select 5 parishes within each division as supervision areas, where we shall randomly select 50% of the villages using a table of random numbers. Within each selected Lot, at least 20 households will be randomly selected for participation.

We are aware that a segment of the population may not be residing in urban centers but working there during the daytime. For such to be considered, we shall list all the households and places of work within the selected urban centers. The places of work will include saloons, bars/restaurants, and markets. In the listing questionnaire, we shall include questions related to the number of people living or working in the dwelling and residence status. For those interviewed at their homes, we shall ask questions on other health and socio-economic variables including the number of people staying in the household, household structure, household assets, and health facility that is usually utilized. For those interviewed at their places of work, questions will include the number of hours spent moving to their workplace, type of business, type of transport, availability of hygiene and sanitation measures, a health facility that is usually used, and workplace sexual harassment perception. We shall also take the household and place of business or workstation geocoordinates. The place of work interviews will only include female respondents that will be working in the selected working stations.

The inclusion criteria are women and men within the reproductive age group, aged 15–49 years for women and 15–54 years for men, residency in the study area and household heads capable of providing informed consent. Those aged less than 18 years but are married (also pregnant or having children) will be considered emancipated minors and will provide individual consent. We shall exclude women and men who have a severe illness at the time of the survey and refuse to consent. The study subpopulation groups are Female youth, 15–24 years, Male youth, 15–24 years, Female adults, 25–49 years, and Male adults, 25–54 years (Table 6). In Each village, we shall interview a fixed number of 20 respondents from each category. The number of interviews in each parish will be divided across the selected villages (Table 6).

The final household data collection tool will include questions related to fertility (birth and pregnancy history), FP use experience, sexuality (age at first sex, recent time of having sex, protected sex), knowledge of FP methods, fertility, and FP intention.

Furthermore, to assess the preference for the package of services and FP options, we shall embed discrete choice questions within the survey questionnaire. The discrete choice approach is based on the assumption that healthcare interventions and services can be described by their attributes and that an individual's valuation depends upon the levels of these attributes. In the discrete choice, the respondents will be asked to choose between two or more alternatives based on a set of attributes and levels. The discrete choice approach will facilitate greater knowledge of the relative importance of the various attributes and the trade-offs that individuals will be willing to make between these attributes. Thus, the discrete choice approach will provide an opportunity to determine societal preferences. In this study, the choice attributes

**Table 6. Population sub-sample and sample size distribution.**

| Population sub-samples | Sample size |
|---|---|
| 1-Female youth, 15–24 years, 2-Male youth, 15–24 years, 3-Female adult, 25–49 years, 4-Male adult, 25–65 years | 5 administrative divisions (2 Iganga and 3 Jinja) X 5 supervisory areas x 20 sets of interviews x 5 population subgroups N = 2000 |

for the preferred package of health services will include cost, FP types (long-lasting and emergencies), distance to the facility, time taken to receive the service, and facility ownership (government, private, and private not-for-profit). The levels of service provision (health facilities, community health workers, community drug shops, and outreaches) will be considered as an alternative.

*Health facility mapping and assessment.* We are aware that the list of the facilities that may be provided by the district health offices or ministry of health may exclude some of the health services centers, in particular, the drug shops and pharmacies. To generate a sampling frame of health facilities, we shall map all the health facilities in the municipalities and group them by levels and administrative authority. During the mapping, we shall take the geocoordinate of the facilities, which we shall use with the health household geocoordinates for distance measurements. The health facility sample was determined based on the following formula.

$$n = \left( (z^2 * p * q) \big/ {}_{[} ME^2 \right] \right) * d \text{ and } S = \left[ {}^{n} \big/ {}_{(} 1 + \{n - 1\}/N) \right]$$

Where;

- S is the final sample size when n is close to the population

- *n* is the sample to be calculated,

- $z^2$ is the square of the normal deviate at the required confidence level,

- *ME* is the margin of error,

- *p* is the anticipated proportion of facilities with the attribute of interest,

- *q* is, and $1-p$

- d is the design effect

- *N* is the total number of the health facilities

Assuming 50% of the health facilities provide FP services (unknown proportion), 5% marginal error (ME), and 95% confidence interval (z = 1.96), the health facility sample size in each district is 385. However, this is higher than the number of health facilities in each district (Table 7). Adjusting the population (number of health facilities) the sample size for Iganga and Jinja municipalities is 80 and 77 health facilities, respectively.

**Stakeholder mapping.** We aim to generate information on stakeholders at family, community and decision/policy levels that affect the availability and accessibility of health services. This information will be collected in different phases. First, through desk review, we shall document all those stakeholders identified in scientific and grey literature. Second, through workshop and project dissemination meetings, we shall ask to involve the participants in outlining and updating the list of the available stakeholders and how influential they are. Lastly, we shall ask questions on the different people that have influence and authority over the availability and accessibility of FP services, which will be quantitatively collected in the household survey and qualitatively collected in in-depth interviews, focus group discussions, and key informant interviews. Nevertheless, we shall have a specific tool and documentation form that we shall use to collect information on stakeholders.

During the meetings, we shall apply interview-based mapping tools such as Net-Map to help us understand, visualize, discuss, and improve situations in which many different actors influence outcomes. More specifically, the stakeholders' mapping will help us determine: what actors are involved in a given network, how they are linked, how influential they are, and what their goals are.

**Table 7. Number of health facilities by level.**

| | Iganga municipality | Jinja municipality |
|---|---|---|
| **Levels** | **Number** | **Number** |
| Hospital | 2 | 2 |
| H/C IV | 0 | 4 |
| H/C III | 2 | 3 |
| HC II | 5 | 8 |
| Total | 8 | 17 |
| **Other health facilities** | | |
| Private pharmacies | 14 | 14 |
| Drug shops | 52 | 20 |
| Clinics/Doctors/Dental | 8 | 39 |
| Allied clinics | 13 | 15 |
| Unlicensed clinics and drug shop | 14 | 8 |

Note that all government health facilities, private not-for-profit, and higher-level private health facilities (clinics and hospitals) will be included.

The health facility assessment will collect information on the availability of FP services including FP supplies, staffing, funding, and protocols. Additionally, we shall collect information on the number of clients that the facilities have served in the last three years before the implementation of the project.

We aim to determine the stakeholders' linkages and levels of influence to understand if we need to strengthen the links to an influential potential supporter (high influence, same goals) and those that help empower or curtail access to the FP services.

## Fieldwork preparation and data management

We shall employ and train 5 teams of data collectors: qualitative and quantitative. Both teams will include those who have had experience in respective data collection methods and understand the local context including the language. The first phase of data collection that will involve field work is the health facility and household mapping. During this phase, we shall collect telephone contacts of all people who are potential respondents. Thereafter we shall collect data using tablets through household visiting. During the interviews, the data will be entered in a predesigned Open Data Toolkit (ODK) form. We shall employ 4 data editors (2 in each district) who will be responsible for reviewing the individual form before it is submitted to the central server. We are aware of the disruption that might be caused by COVID-19 and, thus, in case we find it impossible to conduct face-to-face interviews, using the telephone contacts documented during the household listing, we shall conduct telephone interviews.

## Data analysis

**Qualitative data.** The qualitative data collected before and after the implementation of the intervention will be transcribed and reconciled with notes recorded during the interviews and then analyzed using thematic analysis following the 6 steps recommended by Braun and Clarke (2006) [61]. Our analysis will follow intersectionality theory, which we shall apply to unmask the factors that perpetuate inequities in access to FP.

**Quantitative data.** The quantitative data collected from the health facilities within the study area will be analyzed for FP utilization, and the number of deliveries and postnatal care visits per month. This data will be analyzed for change in the utilization of FP services before, during and after the implementation of the intervention. The data from the health facility assessment will also be analyzed for the availability of an adequate number of trained staff, equipment, FP commodities, drugs and supplies prior to as well as the quality of care before

the start of the intervention and at the end. The data from the household surveys will be analyzed for percentage-specific FP indicators such as the unmet need for FP, contraceptive use and related factors. We shall use propensity score matching from the surveys to measure and ascertain the influence of the intervention on the utilization of health facility services. The data will also be analyzed for behavioral and social indicators within the urban context. The interpretation and presentation of information on each of the lots will be guided by the LQAS tables. Survey weight will be generated for the analysis of coverage indicators.

**Stakeholder analysis.**   First, to identify the actors and how they might impact a project's success, we will identify the influence of individual actors on achieving the project outcome through a desk review and participatory internal meetings or workshops. The identification of actors will not only target those that have power over macro-decision-making but also those that affect the consumers' decision-making to access services. Subsequently, to determine the level of influence and the connection as well as how their influence and connections might impact a project's success, we shall employ Social Network Analysis (SNA) to map the influence of individual actors and their relationships in relation to achieving the project outcome through the same process of data collection. We shall generate participatory SNA graphs that will visually present the stakeholders' influence and connectedness on FP services availability and accessibility. Using visual network software such as UCINET, Kumu and Gephi, we shall analyze the network structural patterns using measures that show the relationships between nodes (tie strength), the key nodes within the network (network centrality), the distance between nodes (degree centrality), the approximate importance of each node (Eigenvector centrality), and the centrality of a node within the network (betweenness centrality).

**Sustainability.**   There will be continuous engagements of all stakeholders to build the capacity to be innovative and use available resources in their communities for improving access to health services. The use of the human-centered design approach to adapt high-impact solutions and other participatory approaches including co-creation will foster sustainability. The study will therefore be designed and implemented to have the implementation learning embedded within the council, district, health facility and Ministry of Health planning. Through the study, strategic engagements will be done to support the institutionalization of the initiatives being implemented. Best practices shall be documented and shared with all stakeholders at all levels within the country and internationally. The key output will be best practice guidelines that increase access to VFP services, communities with the capacity to use available FP services and strategic partnerships with local governments, civil society organizations as well as non-governmental organizations. There will also be discussions with partners at district and national levels to take up the intervention in other areas of the country.

## Discussion

Using the human-centered design approach, we shall be able to develop a tailored package of FP interventions that matches the community context. Our interventions will be tailored to the main domain of known high-impact interventions. Aligned with the global agenda of improving women's health and well-being [5] and Uganda's Family Planning costed implementation plan that calls for continuous research on FP dynamics [21], including what works to improve FP services in different contexts, our process documentation and evaluation approach will address questions related to what package of FP interventions work, for whom, under what circumstances and why in an urban setting.

Guided by strong learning and implementation flexibility, our implementation will provide specific interventions that work for various segments of populations in an urban context. We believe that both the demand for and supply of contraceptives are affected by several barriers

that require a holistic and community-acceptable approach that addresses them. Moreover, these factors affect the groups within the community differently with some segments exposed to multiple barriers. Our approach of working with the municipal and city councils will contribute to the institutionalization of FP interventions within the local government planning and budgeting. Furthermore, our implementation learning will contribute to the identification of interventions that could be replicated in other urban settings within Uganda and other countries within the region that share similar urban features.

## Supporting information

**S1 File. Implementation risks and mitigation.**
(DOCX)

## Author Contributions

**Conceptualization:** Rornald Muhumuza Kananura, Catherine Birabwa, Othman Kakaire, Peter Waiswa.

**Writing – original draft:** Rornald Muhumuza Kananura, Catherine Birabwa, Jacquellyn Nambi Ssanyu, Othman Kakaire, Peter Waiswa.

**Writing – review & editing:** Rornald Muhumuza Kananura, Catherine Birabwa, Jacquellyn Nambi Ssanyu, Felix Kizito, Alexander Kagaha, Sarah Namutanba, Moses Kyangwa, Othman Kakaire, Peter Waiswa.

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
