## [Editor Report · Decision Letter 0]

3 Sep 2023

PONE-D-23-10212Increasing coverage and uptake of voluntary family planning in urban areas of central-eastern Uganda: an implementation research study protocolPLOS ONE

Dear Dr. Kananura 

Thank you for submitting your manuscript to PLOS ONE. After careful consideration, we feel that it has merit but does not fully meet PLOS ONE’s publication criteria as it currently stands. Therefore, we invite you to submit a revised version of the manuscript that addresses the points raised during the review process.

We look forward to receiving your revised manuscript.

Kind regards,

Stella Zawedde-Muyanja, MBChB. MPH

Academic Editor

PLOS ONE

Journal Requirements:

"The implementation of the project is funded by John Templeton Foundation, Grant number 62045.  The funder had no role in the writing of this manuscript and therefore, identified error in writing and reporting is solely the responsibility of the authors. "

Additional Editor Comments:

The authors submitted an implementation research protocol to improve uptake of family planning services in Eastern Uganda. The protocol is well written and describes the study aim, study design and study setting.

The protocol clearly outlines three steps of implementation the formative phase, the implementation phase and the evaluation in line with what is expected of implementation research protocols.

For the formative phase: the protocol describes the methods to be used, people to be interviewed, how they will be selected. The protocol also describes how findings from the formative phase will be incorporated into different phases of the implementation phase.

For the implementation phase, the protocol outlines the different interventions and sample size calculations where necessary . The data management and data analysis plans are well described.

Finally, the protocol outlines how the project will be evaluated and states which theoretical framework will be used for the evaluation.

The status and timelines of the study have been provided and the authors have indicated that data collection has started but is not yet complete

My comments:

a) Specific aims should be reduced to three: As currently stated, Specific aim 2 is a repetition of aim 1 and aim 3 and can be incorporated into those two.

b) The use of both a conceptual framework and a theory of change is repetitive and may be confusing. For implementation, a theory of change may be more relevant. Authors could consider using only the theory of change and incorporating elements currently outlined in the conceptual framework into it.

c) The use of so many frameworks in the same text is confusing for the reader. The authors should an overarching framework for the study: If the human centered design is the overarching framework, then Activities under Table 1 should be incorporated into Table 3 under the subheadings discover, ideation and prototype. For discover phase of the HCD, the data sources for the quantitative data reviews should be mentioned.

d) It is then not clear where the Replicating Effective Programs Framework fits into the study. if this is the main framework for the study, then the authors may consider not mentioning the human centered design.

e) Table 2 and its supporting text (Implementation risk mitigation) is important for programming but may not be critical to the research protocol and may be attached as an annex

f) Authors may consider registering their protocol with a registering body and providing a registration number with the next submission

---

## [Author Response · Author response to Decision Letter 0]

18 Sep 2023

Journal Requirements:

Thank you for the suggestions. We have formatted the document following the provided guidelines.

Thank you for bringing this to our attention. We have duly reviewed the grant number and made the necessary corrections.

"The implementation of the project is funded by John Templeton Foundation, Grant number 62045. The funder had no role in the writing of this manuscript and therefore, identified error in writing and reporting is solely the responsibility of the authors. " Please state what role the funders took in the study. If the funders had no role, please state: ""The funders had no role in study design, data collection and analysis, decision to publish, or preparation of the manuscript."" If this statement is not correct you must amend it as needed. 

Thank you for your suggestion. We have now included the funder's role in the required format as per your recommendation. 

We are still in the process of collecting and cleaning all the required data. This statement was intended for those who may want to use our data in future. We have amended the sentence.

These have been provided before reference list as: 

Figures 

Fig 1: Theoretical framework 

Fig 2: Implementation theory of Change 

Fig 3: Implementation phase 

Fig 4: Indicators and their measurements 

Supporting documents 

S1: Implementation risks and mitigation

References have been revised and they are correct 

Additional Editor Comments:

a) Specific aims should be reduced to three: As currently stated, Specific aim 2 is a repetition of aim 1 and aim 3 and can be incorporated into those two.

We thank the academic editor for the suggestion. We have removed the objective 2. 

b) The use of both a conceptual framework and a theory of change is repetitive and may be confusing. For implementation, a theory of change may be more relevant. Authors could consider using only the theory of change and incorporating elements currently outlined in the conceptual framework into it.

We agree with the academic editor that this was confusing. We have created a section on theoretical framework and have shown how this guided the development of the implementation theory of change.

c) The use of so many frameworks in the same text is confusing for the reader. The authors should an overarching framework for the study: If the human centered design is the overarching framework, then Activities under Table 1 should be incorporated into Table 3 under the subheadings discover, ideation and prototype. For discover phase of the HCD, the data sources for the quantitative data reviews should be mentioned.

As earlier mentioned, we have created a section on theoretical framework. We have left table 1 as it is since it would be important to have the research questions before the methods section. We think this arrangement aligns with the logical flow of information, enabling readers to understand the research objectives before delving into the methodology

d) It is then not clear where the Replicating Effective Programs Framework fits into the study. if this is the main framework for the study, then the authors may consider not mentioning the human centered design.

We thank the academic editor for this suggestion. We agree that this was confusing. Actually, the figure summarises the implementation phases and as such we have revised the text indicating how it aligns the HCD. 

e) Table 2 and its supporting text (Implementation risk mitigation) is important for programming but may not be critical to the research protocol and may be attached as an annex

We thank the academic editor for pointing out this. We have made edits and have attached the table as a supplement. 

f) Authors may consider registering their protocol with a registering body and providing a registration number with the next submission. 

While our protocol cleared by the UNCST after comprehensives review and approval by Makerere University School of public health Institutional review board, we have considered the advice of having it registered on the Open Science Framework (OSF) repository Registries (https://osf.io/vqxu9 ; DOI: 10.17605/OSF.IO/VQXU9).

---

## [Editor Report · Decision Letter 1]

11 Oct 2023

Increasing coverage and uptake of voluntary family planning in Uganda’s emerging municipalities and secondary cities: an implementation research study protocol

PONE-D-23-10212R1

Dear Dr. Kananura,

We’re pleased to inform you that your manuscript has been judged scientifically suitable for publication and will be formally accepted for publication once it meets all outstanding technical requirements.

Kind regards,

Stella Zawedde-Muyanja, MBChB. MPH

Academic Editor

PLOS ONE

Additional Editor Comments (optional):

I thank the authors for taking the time to respond to the comments. This has made the paper easier to read. I request the authors to do one final round of proof reading to make sure the paper is free of clerical errors.

For example

a) in the abstract, Lines 4-5 ..."The implementation of current interventions in a manner that encourage client’s informed choice and to voluntarily use services are currently not well understood" needs correction.

b) in the discussion (abstract). The second sentence is similar to the first and maybe omitted.
---

## [Editor Report · Acceptance letter]

9 Jan 2024

PONE-D-23-10212R1 

PLOS ONE

Dear Dr. Kananura, 

I'm pleased to inform you that your manuscript has been deemed suitable for publication in PLOS ONE. Congratulations! Your manuscript is now being handed over to our production team.

Kind regards, 

on behalf of

Dr. Stella Zawedde-Muyanja 

Academic Editor

PLOS ONE